# Machine Learning-Based Radon Monitoring System

Diego Valcarce [1], Alberto Alvarellos [1,2], Juan Ramón Rabuñal [1,2], Julián Dorado [1,3,4] and Marcos Gestal [1,3,4,5,*]

1    Department of Computer Science and Information Technologies, University of A Coruña, Campus Elviña, 15071 A Coruña, Spain; d.valcarce@udc.es (D.V.); alberto.alvarellos@udc.es (A.A.); juanra@udc.es (J.R.R.); julian@udc.es (J.D.)
2    Centre for Technological Innovation in Building and Civil Engineering (CITEEC), University of A Coruña, Campus Elviña, 15071 A Coruña, Spain
3    CITIC, RNASA-IMEDIR-Group, Faculty of Computer Science, University of A Coruña, Elviña, 15071 A Coruña, Spain
4    Biomedical Research Institute of A Coruña (INIBIC), University Hospital Complex of A Coruña (CHUAC), 15006 A Coruña, Spain
5    IKERDATA S.L., ZITEK, University of the Basque Country UPVEHU, Rectorate Building, 48940 Leioa, Spain
*    Correspondence: mgestal@udc.es; Tel.: +34-981-167-000 (ext. 1379)

**Abstract:** Radon (Rn) is a biological threat to cells due to its radioactivity. It is capable of penetrating the human body and damaging cellular DNA, causing mutations and interfering with cellular dynamics. Human exposure to high concentrations of Rn should, therefore, be minimized. The concentration of radon in a room depends on numerous factors, such as room temperature, humidity level, existence of air currents, natural grounds of the buildings, building structure, etc. It is not always possible to change these factors. In this paper we propose a corrective measure for reducing indoor radon concentrations by introducing clean air into the room through forced ventilation. This cannot be maintained continuously because it generates excessive noise (and costs). Therefore, a system for predicting radon concentrations based on Machine Learning has been developed. Its output activates the fan control system when certain thresholds are reached.

**Keywords:** radon; machine learning; monitoring; applied biosensing

## 1. Introduction

Radon (Rn) is a colorless, odorless and tasteless [1] chemical element with atomic number 86 that was discovered at the end of the 19th century in Germany [2]. Since then, it has been broadly studied, not because of its applications, as it has a short lifetime (almost four days) [3], but because of its radiation. Rn is a biological threat to cells due to its radioactivity. It degrades over time by losing atomic mass, producing radiation and converts into another radioactive element. This cycle is repeated until a stable state is reached, i.e., when radon converts into lead. Alpha particles, beta particles and gamma rays are the types of radiation emitted. Extended exposure to these can be harmful, as this radiation can penetrate the human body and damage cellular DNA, causing mutations and interfering with cellular dynamics, which can lead to the onset of neoplasms.

Rn has been proved to be related to the onset of lung cancer in non-smokers (it is estimated to cause between 3% and 14% of all lung cancers in a country, depending on the national average radon level and smoking prevalence, according to World Health Organization data). Radon has been declared the most frequent cause of lung cancer among non-smokers by the United States Environmental Protection Agency (USEPA) [1] and the International Agency for Research on Cancer (IARC) [4]. Spain—in general—and Galicia in particular, has high natural radon concentrations [5] due to its geology, which is strongly determined by granite, the main source of natural Rn emissions [6–8]. Consequently, it is essential to deploy sensors to detect and quantify the amount of indoor Rn, even at small

concentrations, at least in regions with high natural Rn generation. However, detecting Rn values is just a part of the work because these values will eventually have to be corrected whenever they exceed biologically safe levels to avoid any interaction of Rn with living organisms. The concentration of Rn is measured in terms of its radioactivity, as Curies (Ci) or Becquerels (Bq); both indicate the amount of radioactive material that decays every second (1 Ci = 37 billion, Bq = 37 billion decays/second) [9].

Radon is present in soil, typically in granite rock, which under certain pressures, humidity or temperatures, releases it into the atmosphere.

Actual radon levels in the open air are normally quite low (the average radon activity in the air in the United States is 0.4 pCi/L or 14.8 Bq/m$^3$), but they could vary depending on location and soil geology. Actual levels are also affected by weather variables, such as precipitations and sudden changes in temperature. Radon levels are often higher indoors than outdoors, for example, in homes, schools and corporate buildings. The changes are significant, ranging from 10 Bq/m$^3$ to 10,000 Bq/m$^3$, according to World Health Organization (WHO) data. This organization recommends setting national reference levels at 100 Bq/m$^3$ (2.7 pCi/L), or if this value is not considered feasible due to the characteristics of a certain country, at 300 Bq/m$^3$ (as is the case in Spain, due to its mainly granite soil). Finally, according to these thresholds, the European Union (EU) has defined a maximum yearly exposure level of 300 Bq/m$^3$ as a safe reference level [10].

The enormous interest in society for the control of this gas and its effects on the health of individuals led to an increase in its control and, therefore, in the number of sensors installed for detecting it.

Several studies are available in which radon gas was measured in soil [11–13] and in groundwater [14–16] (sometimes related to earthquakes or volcanic activity [17,18]). Other papers discuss the use of Artificial Neural Networks [19] or other Machine Learning (ML) approaches, but mainly use previously stored data that are then analyzed offline or from a more statistical point of view [20].

This paper aims to discuss a family of Machine Learning models that can predict indoor radon levels knowing the current Rn levels so that indoor radon exposure can be reduced using mechanical devices controlled in real time by intelligent predictions. Section 2 lists the methods and materials used, as well as the data, and Section 3 will present the results obtained. Finally, the models presented will be discussed in Section 2.

### 1.1. Radon Concentration Detection

There are several options for detecting radon concentrations, both in the short and long term. All of them are usually reliable, but it is important to remember that radon concentrations are affected by several elements including, but not limited to, temperature, pressure and/or humidity.

Several devices can be used [21,22], but three approaches or methodologies can be discussed depending on the type of sampling used for measurement: instantaneous, continuous measurement and integrated methods.

### 1.2. Radon Mitigation

However, radon detection is only the first step of the process. Radon concentrations must be measured but, if they exceed certain levels, the second step—radon mitigation—should be started to reduce them to safe levels by means of air renewal in the room or laboratory affected.

There are also different mitigation alternatives [23–25]. These measures need to be taken in enclosed spaces, as these are the most likely places for radon to accumulate. The most common ways of reducing these concentrations are listed below:

- Increasing subfloor ventilation.
- Installing a radon sump system in the basement or under a solid floor to collect the radon and subsequently expel it into the atmosphere.



- Avoiding the passage of radon from the basement into living spaces (e.g., by means of sealing floors and/or walls).
- Improving the ventilation of the building (with fans or crossed ventilation).

This reduction in the radon concentration can be accomplished by opening a window or a door in the room to let the fresh air in to renew the current one (or at least to reduce the concentration by diluting particles in a larger amount of air). In larger spaces, some sort of mechanical aid would be necessary (i.e., air flow systems) to move the required amount of air in the room in a suitable amount of time (typically one hour or less) and replace the contaminated air with clean, fresh air.

Other alternatives for reducing radon levels are available, but they are usually more costly to implement or are aimed at new construction buildings. They include sub-slab depressurization (or active sub-slab suction) systems that pull radon directly beneath the home's foundation and soil to then expel it harmlessly above the roof, where it dissipates very quickly. The quantity and position of the suction pipes required are determined by the ease with which air may flow in the crushed rock or soil beneath the slab, as well as the concentration of the radon. Only one suction point is frequently required. It is the most common and usually the most reliable radon mitigation system; other approaches use pipes that are inserted into a drain tile and expel the soil gases outside. In this case, covers are placed on the sump basket. Another option, usually used in crawl spaces, consists of using submembranes, a plastic sheet to cover the floor that extends up onto the wall and seals the building. Here, a pipe penetrates under plastic sheeting, pulls out the radon and expels it outside. Other approaches use sealing cracks and other openings in the building's foundation when they are visible.

The system proposed in this article employs an air fan to introduce fresh air into the room. The fan is turned on automatically for the intelligent system to keep the radon concentrations within safe values. By making use of a regression model, it seeks to anticipate when the maximum values will be reached in order to turn on the air flow control system and bring those values below a safe threshold. Once the intelligent model establishes that the predictions are below the threshold under the current conditions, the controller turns off the fan. This enables the people that occupy said spaces to avoid increases in radon concentrations.

Passive systems of mitigation can reduce indoor radon levels by more than 50%. When radon ventilation fans are used too, radon levels can be reduced even further.

## 2. Materials and Methods

The first step consisted of acquiring data on radon concentrations. Data were collected using an external sensor [9,26], which also offers an easy and intuitive model for trying to predict the level of Rn using a simple slope. The system was designed to continuously sample radon levels in real time. These measurement values must be averaged over time (typically 10 min to one hour), then extrapolated for a longer period and finally compared with the system's threshold (regulatory limits are legally established for spaces where people are exposed for extended periods of time). According to device characteristics, new data were provided every 10 min as an aggregate of the collisions within that time interval. This interval allowed us to use whole data from device measurements to get a bigger training set and allowed us the detection of tendency changes as soon as possible. The first observation was made on 25 September 2019, at 9:50 a.m., UTC+2, and the system was trained with the raw data until 3 March 2021, at 4:54 p.m. We have registered seven attributes: the timestamp the data are saved on, radon levels ($Bq/m^3$), temperature (°C), relative humidity (%), pressure in millibars (mbar), total volatile organic compounds (tvoc) and state (whether the ventilation is switched on or off).

The equipment used was a sensor based on a pulsed ion chamber, the RD200M [26]. This sensor is broadly used to perform radon measurements [27–30]. The main characteristics of the sensor for the radon concentration measurements are:

- Sensitivity: 0.81 $cph/Bq/m^3$

- Precision: $\pm 10\%$ at 370 Bq/m$^3$
- Measurement range: 7.4 to 3.700 Bq/m$^3$
- Accuracy: $<\pm 10\%$ (min. error $<\pm$ 18.5 mBq/m$^3$)
- Reproducibility: $<\pm 10\%$ at 370 mBq/m$^3$
- Data interval: 10 min update (60 min moving average)

Figure 1 shows the course of radon over time (N.B.: radon values are usually over the threshold) while Figure 2 depicts the remaining variables (except for the ventilation state, which requires a separate analysis). Most of the experimental values comply with the legislation and are coherent with the specific use of the building and its geographic location.

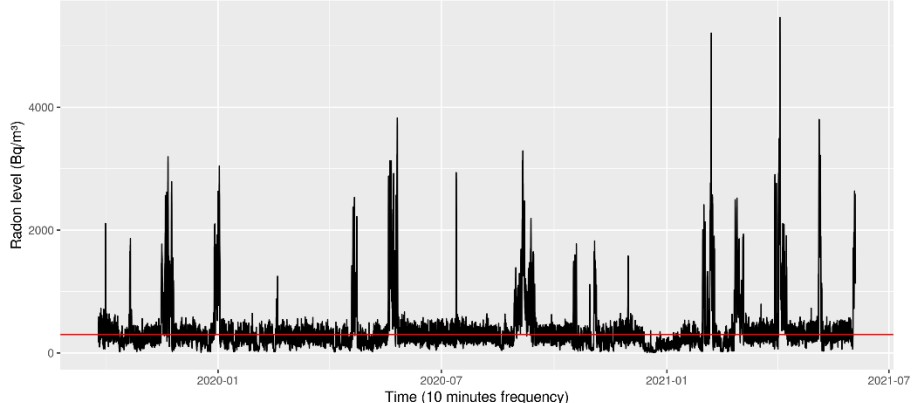

**Figure 1.** Course of radon over time.

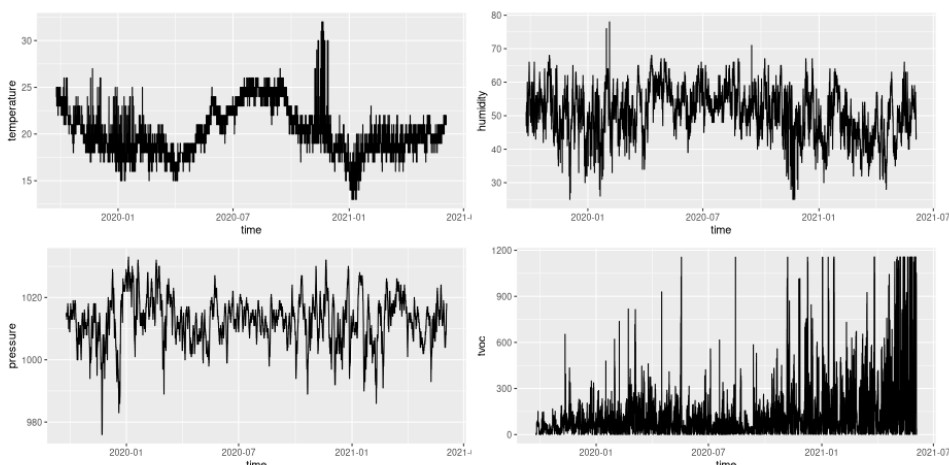

**Figure 2.** Course of the remaining variables over time.

Figure 1 shows that the recommended level (about 300 Bq/m$^3$) is sometimes exceeded; in fact, the mean level is 369 Bq/m$^3$ and the median is 262 Bq/m$^3$. Furthermore, this graph contains an important number of outliers, which can be confirmed in Figure 3. These outliers mainly originate from human mistakes (accidentally disconnecting the control system) and the lack of accurate predictions made by the first prediction model (basically using a slope [9]). Once the final model is selected based on the learning/training step, we will focus on checking the generalization (prediction) ability it can offer. The training was done using the largest "cleanest" signal we found in summer last year, which had around 12,000 samples with almost no outliers.

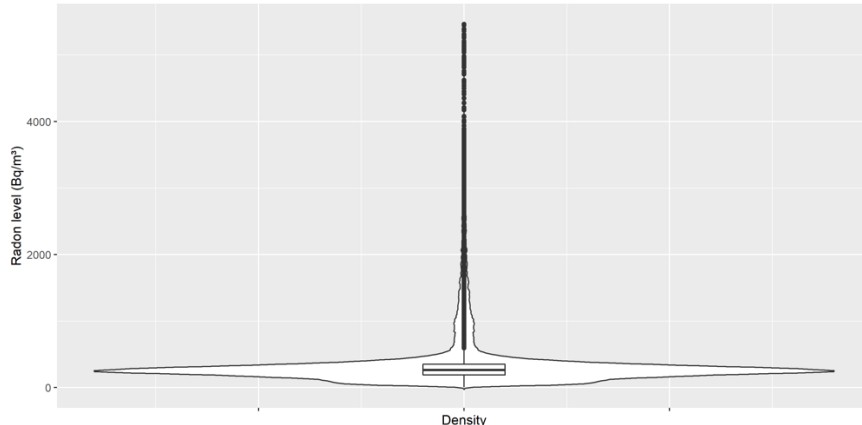

**Figure 3.** Radon violin plot for outlier detection.

The radon concentrations predicted by the intelligent system will be used as input controls for fan control. Nowadays, a simple control function is employed (as shown in Figure 3): when the prediction for the next 10 min exceeds the fixed threshold (now established at 300 Bq/m$^{3[Pl]}$), the controller sends a power-on signal to the ventilation system. Otherwise, when the prediction for radon concentrations goes below this threshold, the controller switches off the ventilation.

Figure 4 summarizes the whole system, highlighting the modules directly related to the control of the ventilation system: the database where the measurements are stored, the container that runs the intelligent model that communicates with the control system of the device by sending alerts and finally the fan itself. Currently, the control device is a simple relay that compares the predicted random value with a threshold (300 Bq/m$^3$) to turn on the ventilation when the prediction is higher and turn it off when it is lower.

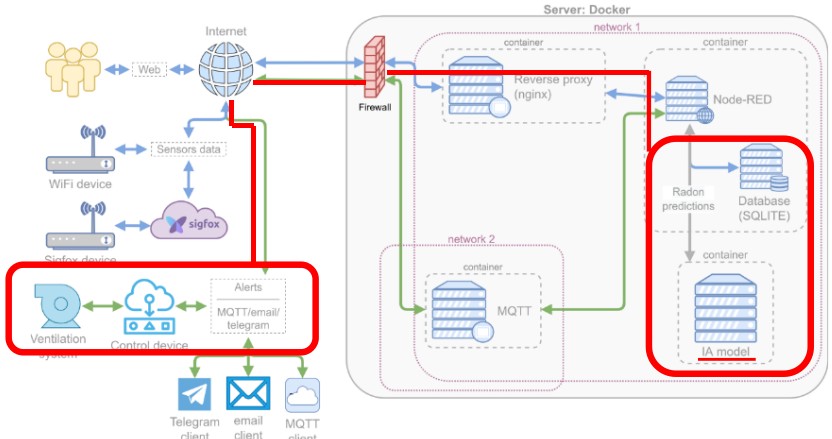

**Figure 4.** General scheme and ventilation system.

As mentioned above, special attention to the ventilation status is required, as it has a direct correlation with the reduction in the amount of Rn within the room [31], as can be seen in Figure 5. This correlation has a $-0.107$ Point-biserial coefficient (used since continuous and binary data are present) with an associated 0 $p$-value, so we can assume that clearly, the ventilation has a positive effect on the radon control

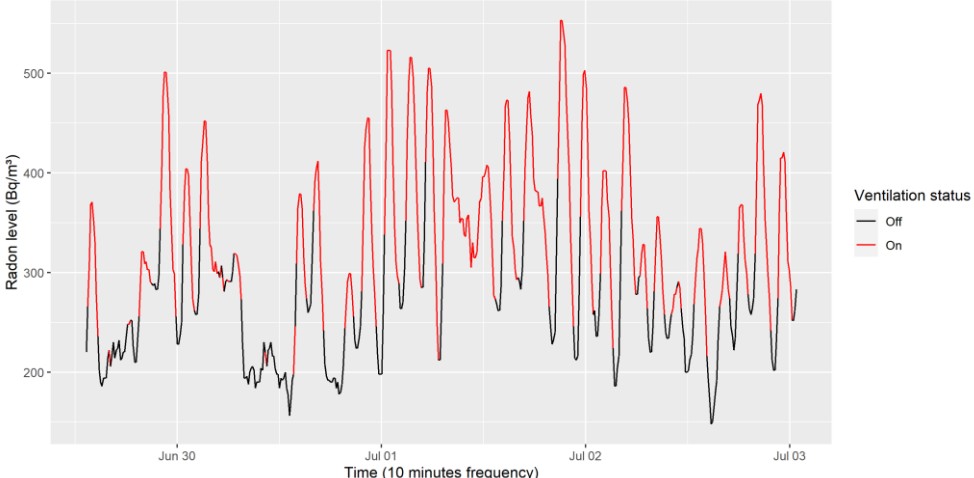

**Figure 5.** Empirical ventilation influences on radon levels.

R language [32] was used for plotting (ggplot2) [33]. Python [34] was applied to build Recurrent Neural Networks (LSTM in this case) using the Tensorflow framework [35]. Both source code and data can be accessed via Github (https://github.com/valcarce01/Radon-prediction, accessed on 1 June 2022).

## 3. Results

The tested RNN models will be discussed in this section. With the aim of comparing the models, a "baseline model" was previously developed. This is a naïve model that predicts the following level of Rn from the current one. This leads to a root mean squared error (RMSE) of 28 Bq/m$^3$ on the selected subsample described in the next subsection.

*RNN*

The RMSE for each model is shown in Table 1; the bold figure represents the best result obtained.

**Table 1.** RMSE for each model built with each combination of covariates (all of them include the radon level) and for each number of previous samples used.

| Covariates/Window Size # | 1 | 5 | 10 | 15 | 25 |
|---|---|---|---|---|---|
| - | 26.57 | 22.19 | 16.45 | 17.21 | 19.33 |
| state | 27.28 | 18.09 | 17.23 | **14.96** | 16.61 |
| humidity | 26.38 | 18.24 | 23.11 | 19.12 | 20.7 |
| pressure | 25.23 | 24.01 | 24.57 | 17.58 | 17.8 |
| tvoc | 23.71 | 17.00 | 17.34 | 17.73 | 18.42 |
| state and humidity | 24.38 | 19.89 | 19.42 | 16.45 | 18.22 |
| state and pressure | 28.15 | 19.48 | 17.5 | 19.95 | 17.01 |
| state and tvoc | 22.93 | 19.95 | 16.09 | 16.13 | 18.35 |
| humidity and pressure | 31.08 | 21.13 | 18.66 | 16.81 | 25.05 |
| humidity and tvoc | 37.52 | 21.30 | 19.84 | 15.77 | 23.16 |
| pressure and tvoc | 28.98 | 34.78 | 18.35 | 16.53 | 19.36 |
| state, humidity and pressure | 29.04 | 24.90 | 18.25 | 16.12 | 20.55 |
| state, humidity and tvoc | 24.47 | 22.59 | 19.35 | 16.87 | 20.1 |
| state, pressure and tvoc | 24.88 | 23.14 | 18.96 | 17.95 | 22.59 |
| humidity, pressure and tvoc | 33.77 | 21.68 | 17.65 | 20.23 | 31.87 |

For the recurrent neural networks (RNN), after several preliminary tests, we found that the best recurrent architecture was based on a set of Long Short Time Memory (LSTM) [20] cells, followed by two dense layers with 16 hidden units and the non-linear Rectified Linear Unit (ReLU) activation function and finally, another dense layer with linear activation (the regressor).

With the goal of connecting the LSTM cells with the first of the dense layers, we encourage the LSTM to predict radon concentrations at each time point, even though the only timestamp that really forecasts the value we are interested in was the last cell. By doing this we introduced new non-linearities (with the dense layers) that involve improvements in both the metrics.

To select the combination of covariates and the window size (the number of samples used as input) that provide the best outcome, we fitted all 75 possibilities (five different window sizes with all the combinations of the covariates). The longer the window size, the more complex the relationship between data can become; we, therefore, gave the LSTM architecture small freedom to increase the number of parameters of the network by selecting the number of hidden LSTM cells as twice the window size. The RMSE in test forecasting for each one of the 75 models is shown in Table 1, where the bold represents the best result obtained.

As expected, the state is the most useful covariate for predicting future concentrations of Rn. It is worth remembering that this is just a first step towards selecting the baseline model to be exploited afterward.

Considering the above results, the prediction of radon concentrations can be successfully achieved using only the previous concentrations and the state variable. So, we used 15 previous observations to develop models with different hidden units in the LSTM layer and different feed forward (ff) network configurations. Results are presented in Table 2. Although most of them do not improve the results shown in Table 1, they belong to the best 1% percentile of the RMSEs of all the different models.

**Table 2.** RMSE for each model built with each combination of the number of feed forward hidden units (in rows) and the number of hidden units in the LSTM layer.

|  | 4 | 8 | 16 | 32 | 64 |
|---|---|---|---|---|---|
| $1 \times 16$ neuron dense layer | 15.21 | 15.05 | 16.08 | 14.90 | 14.95 |
| $2 \times 16$ neuron dense layer | 15.45 | 15.21 | 15.92 | 14.93 | 16.15 |
| $1 \times 32$ neuron dense layer | **14.69** | 14.75 | 15.05 | 15.93 | 15.44 |
| $2 \times 32$ neuron dense layer | 15.63 | 16.10 | 16.67 | 16.46 | 17.38 |

With this in mind, an LSTM model with four units in each cell (which represents the dimensionality of the output introduced by the layer) and a dense layer with 32 neuron units was attempted (see Figure 6, where the dimensionality introduced by the hidden cells is omitted). Figure 7 shows a general reconstruction of the test signal we chose, and Figure 8 provides a closer look at where errors can be seen.

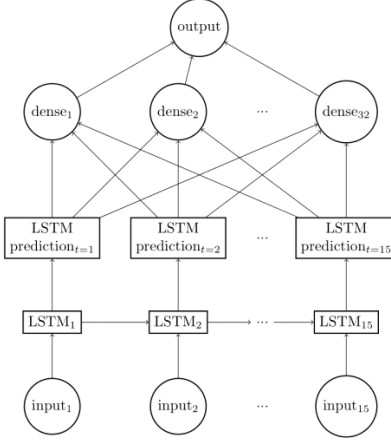

**Figure 6.** Best LSTM model.

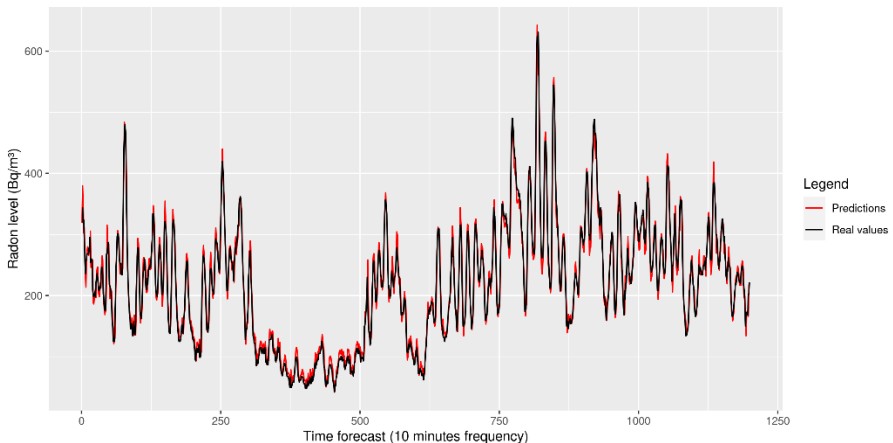

**Figure 7.** Test signal forecasting using the LSTM model presented.

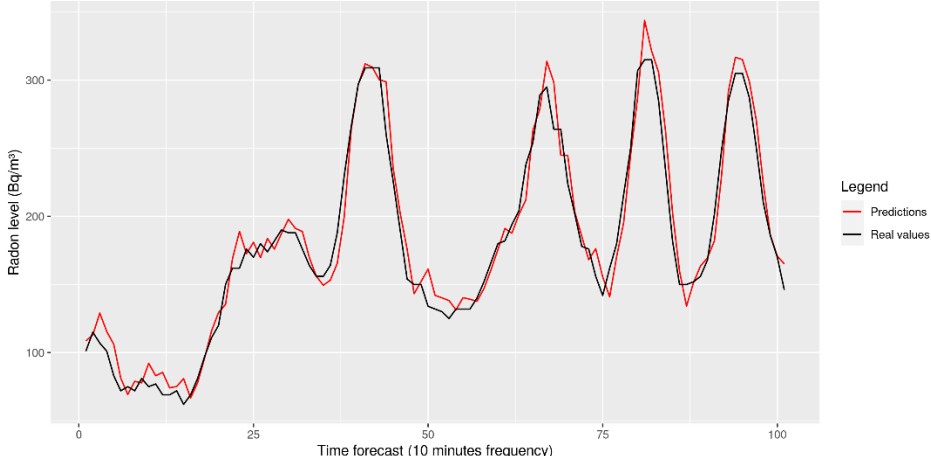

**Figure 8.** Closer look at the LSTM forecasting.

## 4. Discussion

As can be deduced from the models presented above, we are capable of forecasting only at one timestamp (10 min), and this has been a data-driven decision. In Figure 9, we see that the reduction in the level of Rn observed in the samples when ventilation was active for 10, 20 or 30 min is significant and that some conclusions can be drawn: if ventilation was active for 30 min, we cannot say that less Rn is obtained than when ventilation was active for 20 min (despite the lowest values being lower than in the 20 min gap, the highest ones surpass the 20 min ones too). Indeed, with an additional 30 min, the deviation is induced in the model and so fitting it is much harder. However, with a 10-min prediction, we can reduce the level on 75 % of the occasions, which is a nice trade-off in terms of accuracy (very low deviation compared with the other two windows selected) and the electric power required to reduce Rn to the desired levels. This graphic was created from all the data that we have (~1 year), and the conclusion is that the longer the horizon not only has bigger outliers, but also the box plot shows that the 50% of the data (inside the box) has a larger dispersion, so it will be harder to predict an exact value.

Finally, as discussed earlier, the data subsample considered so far was from summer 2020. However, we would like a stable model capable of predicting the values further. To check if the model has the generalization ability, the signal from winter 2021 was considered. It was found that its overall behavior was very similar to that discussed earlier (Figure 10), resulting in an RMSE well below 20, which is even a bit better than that for summer.

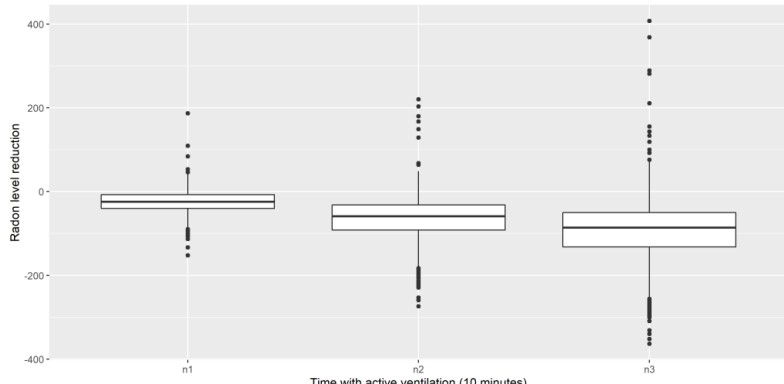

**Figure 9.** Radon level reduction for 10, 20 and 30 min of ventilation.

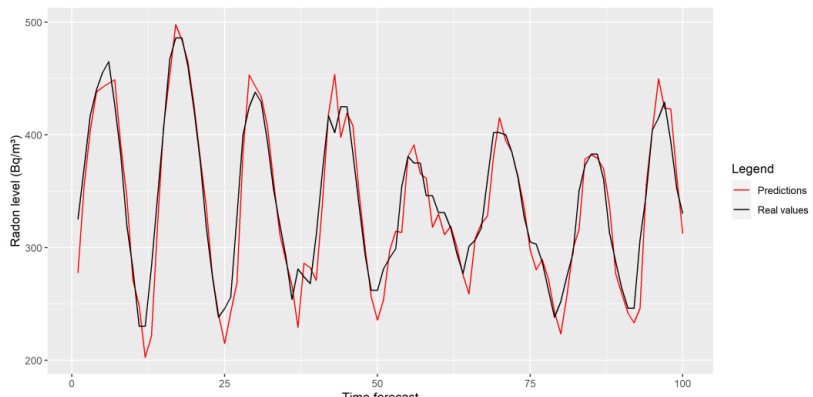

**Figure 10.** Radon forecasting for winter 2021.

Finally, it should be noted that one of the benefits of LSTM is that it is possible to train the models with data from different time points, with no sequence between them. This means that it would be possible to train the model with all the subsamples of "clean" data we have and so, potentially, the difference between the RMSEs from summer and winter would be minimized or eliminated.

The final step consisted of linking the predictor system to the mechanical ventilation system so that it was responsible for managing the engine startup and shutdown. Due to the current configuration of the control module, this should occur whenever the model prediction goes above or below 300 $Bq/m^3$, respectively.

As can be seen, the system is able to maintain radon values within an acceptable range, as shown in Figure 11. This figure represents a small interval from the whole week where the device was making the forecast with very similar results.

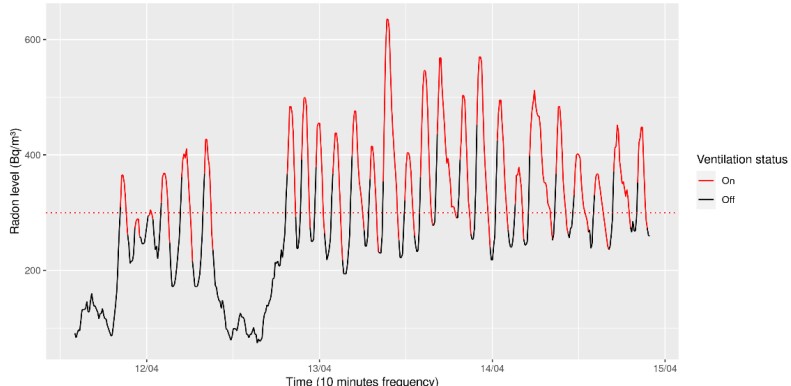

**Figure 11.** Real time operation of the Radon Control System.

Therefore, in our view, the results presented here demonstrate that it is possible to deploy a system to control the levels of Rn indoors by using a mechanical ventilation system controlled by a Machine Learning model.

## 5. Conclusions

The work presented is part of a larger project to monitor radon levels in different rooms. In this study, the feasibility of developing an intelligent system that autonomously switches on and off the fan that is responsible for keeping a room ventilated has been tested. This can be achieved with a simple model. In fact, of all the measurements taken, it is enough to use the time series of radon measurements to make correct predictions, with the other measurements of humidity, etc., providing very little useful information. This simplicity is also helped by the fact that all measurements are performed in the same room, which makes the signal concentrations very repetitive over time. Work is already underway to obtain data from different rooms/locations to solve this problem.

It is not possible to obtain better results precisely because of this simplicity, introduced by the type of controller used. More tests need to be performed with controllers that allow more interaction, different on/off thresholds or that allow the control of the operation in terms of increments/decrements of measurements between consecutive intervals, etc.

Due to the characteristics of the current installation, it has not been possible to test the system under interference conditions either. In future versions, attention should be paid to the help provided by multisensor systems or e-noses [36,37] in this field.

**Author Contributions:** Conceptualization, M.G.; methodology, M.G., J.D. and J.R.R.; software, A.A. and D.V.; validation, J.D., J.R.R. and M.G.; formal analysis, J.D and M.G.; investigation, J.R.R., M.G. and J.D.; resources, M.G and J.R.R.; data curation, A.A. and M.G.; writing—original draft preparation, D.V. and M.G.; writing—review and editing, D.V., M.G. and J.D.; visualization, A.A. and D.V.; supervision, M.G., J.D. and J.R.R.; project administration, M.G.; funding acquisition, M.G. All authors have read and agreed to the published version of the manuscript.

**Funding:** This work is supported by Instituto de Salud Carlos III, grant number PI17/01826 Collaborative Project in Genomic Data Integration (CICLOGEN) funded by the Instituto de Salud Carlos III from the Spanish National Plan for Scientific and Technical Research and Innovation 2013–2016 and the European Regional Development Funds (FEDER)— "A way to build Europe." This project was also supported by the General Directorate of Culture, Education and University Management of the Xunta de Galicia ED431D 2017/16, the "Drug Discovery Galician Network" Ref. ED431G/01, and the "Galician Network for Colorectal Cancer Research" (Ref. ED431D 2017/23). This work was also funded by the grant for the consolidation and structuring of competitive research units (ED431C 2018/49) from the General Directorate of Culture, Education and University Management of the Xunta de Galicia, and the CYTED network (PCI2018 093284) funded by the Spanish Ministry of Innovation and Science. This project was also supported by the General Directorate of Culture, Education and University Management of the Xunta de Galicia "PRACTICUM DIRECT" Ref. IN845D-2020/03.

**Institutional Review Board Statement:** Not applicable.

**Informed Consent Statement:** Not applicable.

**Data Availability Statement:** Not applicable.

**Conflicts of Interest:** The authors declare no conflict of interest.

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
