# Peer review of "Machine Learning-Based Radon Monitoring System"

_chemosensors, doi:10.3390/chemosensors10070239_

Round 1
Reviewer 1 Report
The manuscript could be more interesting if the authors dealt better with the general part relating to Radon and its measurement.
The introduction should contain references to works that have dealt with similar themes and 7 or that have used the same models of analysis, see for example Ambrosino et al., (2020). Analysis of 7-years Radon time series at Campi Flegrei area (Naples, Italy) using artificial neural network method. Appl. Radiat. Isot. 163, 109239.
https://doi.org/10.1016/j.apradiso.2020.109239
Others also deal with this theme.
For the treated topic it is not necessary to report the possible measurement techniques (see Radon concentration detection ) and actions to reduce Radon in buildings (see Radon mitigation ). They are treated better elsewhere; if it is considered useful, just report some references.
Some parts of the introduction and Radon mitigation are the same as those of the ref [4].
Otherwise, from this manuscript the instrument of measurement with which the data that are analyzed were obtained is not even named. Reference is made to [4], where reference is made to ref [18] which contains a link to an inactive website.
Regarding this unknown instrument, they said that it can measure in intervals of 1 to 20 min. Let them understand how this instrument works, since those who use the decay products of Radon cannot have a measurement time of less than 20 min.
I believe that considering measurement and analysis intervals of 10 min does not make any sense.
They say that the measurement range starts from 0.1 Bq / m3, while in ref [4] it starts from 7.4 Bq / m3. But, are you talking about the same tool?
If the instrument provides 0.81 cph at 1Bq / m3, how many days of measurement are required to have an error of 10%?
Correctly indicate the measurement errors based on the calibration error of the instrument and the measurement time.
In line 177, the average value is reported even with two decimal places. Non deviation standard is reported.
Continuous reading methods should be Continuous measurement methods.
Most of the captions of the figures do not make explicit the content of the figure. Also, the units of measurement of the horizontal axis is not always reported or clear.
Advanced signal analysis methods are used but a qualitative correlation is shown in Figure 4. What quantitative significance does it have?
How many analyzes do the data in Figure 7 come from? The mean value of the box plot shows an increase in the reduction but there are also many outliers. Perhaps, it would be better to show it in another way.
It is not clear how the ARIMA model was applied, but since it does not work well with this data it seems pointless to report it.
The methods of application and the results of the LSTM model should be better explained in terms of the physical parameters involved.
Specify for good forecasting with this method on how long the model has been based and how long the forecast can be made.
Reviewer 2 Report
The paper presents a machine-learning based radon monitoring system. Although the topic is quite interesting and very important for human health, the paper lacks the quality to be published in its present form. I have the following comments about the manuscript.
1. Although the introduction is quite long, there are no references about other works. The Aurhors only refer a previous work of their research group, they must perform literature revision about other similar works.
2. The Authors do not mention the kind of sensor that was used in the study. I think it is the same as the one used in their previous work, but they have to write it in this paper too.
3. There are not a simple sketch of the ventilation system.
4. Some graphs does not have the units in the axis, for instance Figure 5 and 6, although we know that the units are minutes, it is necessary to write them. Please correct.
5. I suggest in figure 1 to put a line in the recommended level (around 300 Bq/m2)
6. The obtained models in the RNN must be included.
7. Please extend the discussion and include a Conclusions section.
8. Is it not clear how the ventilation system was controlled.
9. In line 272, the authors claim "that it is possible to train the models with data of different time steps, with no sequence between them". I think that it is necessary to train the models with such kind of data in order to obtain more robust results.
10. Finally, I think the entire manuscript must be checked by a native speaker to enhance the English writing. Even in the abstract there is a misstype besides other gramatical problems.
Reviewer 3 Report
The authors discuss a system to monitor/to manage an indoor radon concentration. This work follows previous authors’ fundings published in Sensors 20 (2020) 752. Such measurements are rather rare in literature and the obtained knowledge is worth to publish.
There are two major concerns regarding the content.
1. The authors do not deliver the proper information on the radon sensor employed in the study (except for referencing the previous paper, line 151). Still, it should be described. Moreover, the authors note in Abstract that “The Radon concentration in a room depends on multiple factors: ambient temperature, humidity level… “. However, they do not deliver results on performing the system under interferences. So, they have to elaborate this issue in Discussion and apply the readers’ attention to multisensor systems capable to help with such interferences (see, for instance, Sensors 18 (2018) 550 or IOP Conference Series: Materials Science and Engineering 932(2020) 012082).
2. The text abounds by numerous unclear phrases. See, for instance (not a full list):
- line 51: However, detecting the Rn values is just but a part of the work because (eventually) they have to be corrected whenever they exceed the values considered as biologically safe, to avoid the interaction of Rn with living organisms.
- line 66: This organization recommends setting national reference levels 100 Bq/m³ (2.7 pCi/L), or if this value is not considered feasible due to is not considered feasible due to the particularities of the country, 300 Bq/m³ (as is the case in Spain, due to its mainly granitic soil);
- line 79: There are several options for detecting radion concentrations, both short and long term.
- line 104: But, detection is only the first step of the process. Radon concentration must be measured but, if it raises some level, mitigation should be start to reduce it to secure levels so we should renew the air that the room, lab… contains.
- line 118: This reduction can be accomplished by opening a window or door and letting the fresh air to enter in the space to replace de current one (or at least reduce the concentration due dilution of particles in a bigger amount of air).
- line 218: To select the best model all the combinations of the covariates we have were trained, and for each of them, we tried different window sizes to assess how many previous observations are needed to get an acceptable accuracy.
So, heavy English polishing is required.
In addition, the authors are advised to enlarge the scripts at the figures for clarity.
Round 2
Reviewer 1 Report
Radon concentration detection section must be eliminated because it is not needed as long as it is treated later.
Instead, it is necessary to specify whether it makes sense to acquire data every 10 min from the RD200 instrument. I understand that they can be read with high frequency, but what sense does it make based on the operating principle of the instrument?
The ARIMA method is still mentioned in some places in the manuscript.
Bq / m3 is misspelled in some places (Bq / m2 or Beq / m3).
The characteristics of the RD200 are expressed in Bq / m3 and pC / L (lines 168-174) Express all in Bq / m3.
Line 165. The two decimal places of 369.43 make no sense.
Reviewer 2 Report
Although the revised version has been improved, there are still some details in the manuscript. For instance, in line 214 it is written "ontinuous". A thorough revision of all the manuscript is necessary.
After such revision, the paper can be published.
